# Organic Farming as a Strategy to Reduce Carbon Footprint in Dehesa Agroecosystems: A Case Study Comparing Different Livestock Products

**DOI:** 10.3390/ani10010162

**Published:** 2020-01-17

**Authors:** Andrés Horrillo, Paula Gaspar, Miguel Escribano

**Affiliations:** 1Department of Animal Production and Food Science, School of Agricultural Engineering, University of Extremadura, Avda. Adolfo Suarez, s/n, 06007 Badajoz, Spain; andreshg@unex.es; 2Department of Animal Production and Food Science, Faculty of Veterinary Medicine, University of Extremadura, Campus Universitario, 10003 Caceres, Spain; mescriba@unex.es

**Keywords:** organic livestock, extensive management, carbon footprint, life cycle assessment, carbon sequestration, dehesa

## Abstract

**Simple Summary:**

This paper attempts to analyze the impact of organic livestock farming in dehesas through the analysis and review of the carbon footprint of seven extensive organic farming systems in various dehesas in the southwest of Spain. The method used was life cycle assessment, taking into account both greenhouse emissions and carbon sequestration. Greenhouse emissions estimated are those derived from livestock digestion, manure management, soil management, and off-farm inputs (feeding, fuels, and electricity). Carbon sequestration calculations consider carbon fixation due to pasture and crop waste and carbon fixation in soil due to manure fertilization. The farms under study represent all the species bred in the farms and all the habitual farming systems existing in dehesas, with the following types being under analysis: beef cattle, sheep for meat, Iberian pigs, and dairy goats. The emissions identified in the farms under study have been found to be lower than those from conventional farms, with values of 16.27 and 10.43 kg CO_2_eq/kg of live weight for beef cattle, 13.24 and 11.42 kg CO_2_eq/kg of live weight for sheep, 1.19 kg CO_2_eq/kg of milk for goats, and 4.16 and 2.94 kg CO_2_eq/kg of live weight for pigs. The levels of carbon sequestration are also noticeably higher, with compensation being up to 89% in meat producing ruminants’ farms, 100% in dairy goats’ farms, and values compensating the total emissions in the case of Iberian montanera pig farms.

**Abstract:**

This study employs life cycle assessment (LCA) for the calculation of the balance (emissions minus sequestration) of greenhouse gas emissions (GHG) in the organic livestock production systems of dehesas in the southwest region of Spain. European organic production standards regulate these systems. As well as calculating the system’s emissions, this method also takes into account the soil carbon sequestration values. In this sense, the study of carbon sequestration in organic systems is of great interest from a legislation viewpoint. The results reveal that the farms producing meat cattle with calves sold at weaning age provide the highest levels of carbon footprint (16.27 kg of carbon dioxide equivalent (CO_2_eq)/kg of live weight), whereas the farms with the lowest levels of carbon emissions are montanera pig and semi-extensive dairy goat farms, i.e., 4.16 and 2.94 kg CO_2_eq/kg of live weight and 1.19 CO_2_eq/kg of fat and protein corrected milk (FPCM), respectively. Enteric fermentation represents 42.8% and 79.9% of the total emissions of ruminants’ farms. However, in pig farms, the highest percentage of the emissions derives from manure management (36.5%–42.9%) and animal feed (31%–37.7%). The soil sequestration level has been seen to range between 419.7 and 576.4 kg CO_2_eq/ha/year, which represents a considerable compensation of carbon emissions. It should be noted that these systems cannot be compared with other more intensive systems in terms of product units and therefore, the carbon footprint values of *dehesa* organic systems must always be associated to the territory.

## 1. Introduction

Dehesa, situated in the southwest of the Iberian Peninsula, is one of the largest managed agroecosystems in Europe. However, their current environmental situation is alarming, with natural resources, such as the soil, water, and biodiversity, being under great pressure. In spite of this, livestock farming and agriculture can highly contribute to their preservation, although it can be the cause of their accelerated deterioration [1], unless management of the systems is also adequate.

With the increase of the food demand and climate change as the main actors, the dehesa ecosystem will be required to adapt to an increasing lack of natural resources and the reduction of greenhouse gas emissions (GHG) [2]. GHG emissions and climate change represent two of the world’s greatest environmental concerns, with the reduction of GHG emissions being one of the main challenges the European farming industry will face in the forthcoming years.

The fight against climate change has become a current main concern. In this sense, measuring the impact of farming and the agricultural activities of the extensive systems and specifically, of the dehesa areas, is a major objective, as there are major differences between the more extensive and organic production systems and the more intensified systems, which use less natural resources and more animal feeds. These systems are a priori, more sustainable, since they could also generate added value from an economic and environmental point of view [3]. 

In this context, the proliferation of studies on farming GHG emissions provides many well-founded opinions. Papers such as that by Smith et al. [4] indicate that the conversion to organic farming in this specific area would reduce GHG emissions, although it would also reduce production, which would require other areas to increase production in order to offset the lack of supply, and net emissions would therefore become higher. Other papers compare organic sustainable production systems and high-performance farming with the purpose of meeting the increasing food demand, with the conclusion that high-performance farming is as sustainable as organic farming and the choice of system will be fundamental for the future of biodiversity [5].

Other papers such as that of Muller et al. [6] propose organic farming as an essential part of the future of the food systems, together with a dramatic change in the food culture and a reduction in food waste. Reports such as Research Institute of Organic Agriculture (FIBL) and International Federation of Organic Agriculture Movements (IFOAM EU) [7] highlight the contribution of organic farming to the mitigation and adaptation to climate change, pointing out that a future scenario where organic farming increased by 50% in 2030, would yield a potential reduction of 12%–14% in the GHG emissions from the farming industry in the European Union. Such changes would derive from the increase in the soil’s organic matter and a reduction in the use of mineral fertilizers.

For such temporal framework, the southwest of the Iberian Peninsula will be required to accept the coexistence of multiple production models, where organic farming must take part as an alternative to the other models. But, can organic farming production in such ecosystems be one of the strategies to mitigate climate change? 

Although the GHG emissions deriving from farming systems are complex and heterogeneous, the management system proposed by organic farming based on the simplification and adoption of certain practices leading to improving pastures and soils, can mitigate the GHG emissions of the farming systems [2,8].

Several methods can be used to calculate the carbon footprint (CF) of the various production systems, although one of the most popular and internationally-recognized ones is the life cycle assessment (LCA) [9]. Recent papers such as that of Gutiérrez-Peña et al. [10] which analyzes dairy goat farms in the south of Spain, that of Eldesouky et al. [3] which analyzes cattle and sheep farms in the southeast of Spain, or one analyzing the dairy cattle farms in the north of Spain by Nova et al. [11] are some of the examples. Such papers focus on conventional production farms, whereas the present paper measures the CF in organic extensive farms.

One of the main problems when comparing GHG emissions between different livestock production systems is the use and implementation of different methodologies, as well as the level of variation generated by the different emission factors considered. Emission factors provided by default by the Intergovernmental Panel on Climate Change (IPCC) (2006) generate a high level of uncertainty compared to others that are more local or directly measured on-site. In addition to these factors, different results can be found from the allocation of global warming potentials (CH_4_ and N_2_O) and the system limits established. The results may vary if the limits of the system remain on the farm itself or, as in our case, to the entire life cycle of the inputs (harvesting, transport, manufacturing, etc.). Similarly, results may differ depending on the functional units considered, e.g., it would seem clear that measuring the CF per unit of product (kg or L) is less appropriate than doing so per farm area (ha) in extensive systems. For that reason, it is necessary to incorporate carbon sequestration in the GHG emissions balance when we study extensive livestock systems.

Within this context, it becomes necessary to approach a detailed study of the GHG emissions deriving from the rearing of organic cattle, specifically the one reared in dehesas. Such analysis will be performed by species, providing details of the origin of the carbon footprint generated by each aspect associated to production, with the purpose of determining its contribution to the global carbon footprint and establishing the possibility of proposing this sustainable farming model as an environmentally-friendly alternative against the increasing industrialization of this segment.

The dehesa of the southwest of Spain represents over a million of hectares [12] and comprises various farming systems. This paper will focus on the organic livestock production system. The purpose of this paper is to estimate the balance of GHG emissions and CF in seven ruminants and Iberian pig organic farms taking also into account their carbon sequestration potential.

And lastly, the reduction of the carbon footprint is also closely associated with the increase in the efficiency of the production system and, therefore, its profitability [13,14]. This is the reason why future research should include carbon footprint in a system in order not only to improve system sustainability, but also to financially reward the reduction of GHG emissions.

## 2. Materials and Methods

Life cycle assessment (LCA) is one of the methods most frequency used to calculate the balance of greenhouse gas emissions (GHG) in livestock farms, as it is a standard and internationally-accepted means to effectively quantify the environmental impact of a product, and also allows to take into account carbon sequestration [3,9,15,16]. This was the basis for its selection as the most appropriate method for this study. The calculation of the carbon footprint was performed following the UNE-EN-ISO 2006 standards [17,18], the IPCC guidelines [19] for national GHG emissions and their subsequent amendments [20,21], the atmospheric emissions national inventories [22], and also an adaptation of the technique to the Spanish Ministry of Agriculture’s method for the characteristics of the areas under analysis [23].

### 2.1. Case Studies Selection and Data Collection

This research is based on a case studies methodology developed by Yin (1984) [24] on his work titled “Case Study Research: Design and Methods” and it is mainly characterized by an intensive approach to an object of study or unit. It is used for the description of real situations and is applicable, for example, to problems related to the management of enterprises, being in the case of this research, the livestock enterprise as the unit of study.

The farming system under study in this paper can be considered unique: an agro-ecosystem grazed by different livestock species under extensive conditions and giving rise to different products depending on the management that the owners of the farms decide to adopt. All these farms are management units subject to the same soil, climate, and socio-political conditions located in the Spanish region of Extremadura, an administrative unit of governance.

The selection of seven organic farms for in-depth study has been considered as the appropriate method for achieving the objectives of the study, since each of them is characteristic of a representative management system in the region of Extremadura (the regional area on which the study is focused). It should be mentioned that the number of organic farms in the region is very low, in fact, in the case of organic pig and goat farms, the region only has three farms registered in 2017 and they were all included in the study. 

In the recent literature, there are numerous studies that use the case study approach for the analysis of livestock farm management from both environmental and technical–economic approaches, for example, Bernués et al. [25] study the environmental impact and ecosystem services of sheep in Spain, Vellenga et al. [26] compare the use of conventional and organic beef cattle water, and Eldesouky et al. [3] analyze the carbon footprint in dehesa farms in Spain. Works with a technical–economic bias are for example those of Neira et al. [27], Asai et al. [28], and Regan et al. [29].

Data were collected from each of the seven farms by way of one-to-one interviews with the farmers or proprietors of the farms during the first semester of 2018.

### 2.2. Features of the Seven Production Systems

In Table 1 we can see the main characteristics and technical indicators of the seven case studies. The data refers to year 2017.

### 2.3. System Boundaries and Functional Unit

The scope of this study covers the entire process until the finished product, which will vary subject to the type of farm. The limits selected for the organic systems included all the on-farm and the off-farm emissions, understanding them as a dynamic set of activities. The on-farm emissions are all the emissions caused by the cattle (enteric fermentation, CH_4_), manure and soil management, and (CH_4_ y NO_2_). Off-farm emissions are emissions associated with the manufacture and transport of feed for the cattle, the use of fuel, electricity, transport, etc.

Emissions are indicated in two functional units: the first one uses the main type of product in each system, i.e., the kg of live weight per sold animal (in meat farms) and the kg of fat and protein corrected milk (Fat and protein corrected milk (FPCM) in dairy farms) [31] and the second one is based on 1 ha of the total hectares of the farm.

### 2.4. Estimation of GHG Emissions and CF Level in Farms

The method used for the estimation of the GHG emissions is the guidelines established by IPCC for the national GHG inventories [19]. All the emissions are expressed in kg CO_2_eq depending on their potential global warming. These global warming potentials proposed by [20,21] are 1 for CO_2_, 25 for CH_4_, and 298 for N_2_O.

#### 2.4.1. On-Farm Emissions

In order to estimate the on-farm emissions, the following have been taken into account: enteric fermentation, manure management, and soil management. The emission factors were taken from the National Greenhouse Gases Inventory for agricultural processes. Additionally, the existence of more specific emission factors, according to the type of farm and location, provided the opportunity of adapting the methodology and introducing more specific emission factors to the features of the areas under analysis, as well as the manure and soil management [23,32].

As can be seen in Table 2, different emission factors have been used for GHG estimation, choosing local emission factors and/or their adaptation to dryland pasture systems whenever possible. The objective has been to be as close as possible to a Tier 3 level. In this sense, for example, the factors used in the Spanish national inventories are at a Tier 2 or 3 level. This objective has been met in the on-farm emission factors; however, in the off-farm emission factors (system inputs), different sources have been used and in some of the cases, were more distant from the Tier 3 objective.

#### 2.4.2. Off-Farm Emissions

The emission factors of the inputs brought onto the farms were obtained from Bochu et al. [32] and the Spanish National Commission for Markets and Competition [33]. As all of them are organic products, the emission factors were recalculated from an estimate of the factors proposed by Bochu et al. [32]. These factors were calculated by discounting the emissions attributed to transport. In order to calculate this proportion, the ReCiPe 2016 Midpoint (H) V1 [34] method was used with the Agri-footprint mass allocation [35] and Ecoinvent 3 allocation [36] databases.

In terms of fuel emissions, both the emissions generated and the combustion emissions were taken into account. The electricity used in these types of farms is mainly for lighting purposes.

The main emission factors used by species are shown in Table 2.

### 2.5. Carbon Sequestration in LCA

The carbon sequestration concept refers to the changes in the carbon (C) composition levels of the soil. Such changes take place in the soil due to the addition of manure, crop, and grassland waste. Therefore, the C-level composition of the soil can be impacted by the changes in the use of the land and the various management systems applied to the farm.

In terms of methodology, there are several methods that can be used to estimate carbon sequestration. For example, IPCC [19] estimates the changes in soil C levels according to inventories and with a 20-year time horizon. For the purposes of this piece of research, the balance of net carbon flows in the livestock–manure–grassland system proposed by Petersen et al. [38] was used with some variants and by adaptation to other systems of similar characteristics to the systems under study [39]. The main difference with IPCC [19] is the recommendation of using a 100-year perspective in order to analyze the changes taking place in the soil carbon levels in time [39]. Therefore, it has been estimated that 10% of the C added to the soil will be sequestrated in a 100-year time horizon [38]. Another correction introduced in the method was the consideration of crops in the livestock–manure–grassland systems, separately assessing C sequestration according to land use in the farms. In this regard, the calculation of C sequestration in the production systems under analysis is performed by taking into account carbon fixation in airborne and underground pasture waste, crop airborne and underground waste, and carbon fixation from manure and the soil fertilized by it.

Specifically, in extensive organic farming systems, the pasturelands and crop lands can be considered as a form of carbon sequestration and a way to mitigate the carbon footprint these types of production systems cause [3,15,19,40,41]. When we talk about crop lands, we generally mean cultivated meadows or rainfed crops for animal feeding. This is when we consider the residues for the carbon fixation of in the soil.

As some authors have pointed out, when considering carbon sequestration in soil, CF in extensive farms is lower than in intensive farms. In this context, trees play an important role in the carbon cycle and therefore the quantification of the balance between carbon emission and sequestration is one of the main challenges. This way, maximizing carbon sequestration can become a management objective in both agroforestry and rangelands systems [3]. No information is available on annual sequestration due to trees in these systems, so this aspect has not been considered in this document.

## 3. Results

In this section of our paper, we describe the results obtained from our CF calculation. The features of the farms under analysis are shown (Table 3) in the first place. And in the second place, the composition of the emissions according to the various greenhouse gases is analyzed (Table 4 and Table 5).

The results are broken down by emission type, the livestock species of the farm, and its contribution to the carbon footprint, expressed in kg CO_2_eq, kg of CO_2_eq per functional unit, and kg of CO_2_eq per hectare of total farm area.

### 3.1. Technical Features of the Farms under Analysis

Table 3 shows the most significant features of the farms under analysis and their technical–financial indexes with the purpose of contextualizing the results of the CF analysis which will be shown at a later stage. The data has been organized by livestock species.

In Table 3, the farms under analysis are seven different farms with four different livestock species and resources that are adapted to their production models. They all (organic) have a common feature [42,43,44]: all animals are reared and fed in freedom, with the majority of their time spent grazing in the dehesas or pasturelands of the farms.

In these farms, the land type varies according to the geographical area where they are situated: the beef cattle farms are situated in dehesa areas, that is, they include a variable number of holm oaks or cork oaks, as is the case of pig farms. The latter are also associated to these ecosystems due to the end stage of pig fattening, where pigs feed mainly on acorns from the holm oak or the cork oak. These two predominant species (*Quercus ilex* and *Quercus suber*) make up 60% of the national fruit of the montanera system [45]. Additionally, these lands are also used by cattle as pasturelands. On the other hand, sheep farms are located in pasturelands, where trees are scarce in the plains and only exist in the mountain areas, where they share habitat with all kinds of endemic bushes. Lastly, dairy goat farms are situated in areas combining mountain and dehesa as well as crop lands.

In relation to the characteristics of the soil (type/conditions of the farms), these dehesas and pasturelands are mostly acidic soils with low organic matter content and a semi-arid Mediterranean climate. Regarding climate conditions, the predominance is the dry climate with low rainfall and extreme temperatures in the winter and summer seasons.

In terms of livestock stocking rate, beef cattle represent 0.59 LU/ha for weaned cattle and 0.3 LU/ha for fattening cattle, which coincide with the findings of Horrillo et al. [46] and notably less with the research carried out on conventional cattle farms (0.73 LU/ha) by Maroto-Molina et al. [47].

In terms of sheep, the stocking rate is 0.44 LU/ha for farms selling animals at 23 kg and 0.6 LU/ha for farms selling animals at 18.5 kg, in line with the farms with low stocking rates described in papers such as those of Gaspar et al. [48]. The dairy goat farms have a stocking rate of 0.24 LU/ha and in pig farms, the stocking rate does not exceed 0.2 LU/ha.

With regards of the inputs brought into the farms, there are clear differences between the feeding expenses by hectare in ruminant farms (beef cattle–sheep) dedicated to meat production and those of dairy goat and Iberian pig farms. Energy use (fuel and electricity) reveal similar levels, except for one, the farms selling sheep at 18.5 kg. The use of fuels in these farms is attributable to the use of vehicles for employees to move about and the machinery employed in the farming activities. Goat farms, as Table 3 shows, do not have electricity expenses, as all its premises, milking, and refrigeration units, etc. are supplied with renewable energy (solar panels).

Regarding production indicators or farm outputs, some relevant data are: the calves sold per cow, sale of weaned animals (0.73), and sale of fattened animals (0.68), in line with the research carried out by Escribano [49,50]. Another indicator to be highlighted is the weight of the animals sold in each farm, as this allows to identify the differences amongst the production models for each livestock species. For example, the sale of weaned calves or yearlings, the sale of sheep at 18.5 kg or 23 kg in weight, the sale of kids for the purposes of milking the goats, or the sale of 1–2-year-old pigs and fattening montanera pigs in Iberian pig farms.

### 3.2. Greenhouse Gas Emissions

Table 4 includes the contribution of the various GHG in the seven systems under analysis expressed in kg CO_2_eq/FU. It also includes the percentage contribution of the various production processes.

Table 4 shows the dairy goat farm as having the lowest CF levels per functional unit (FU) (1.19 kg CO_2_eq/kg of corrected milk), followed by the Iberian pig dehesa farms (2.9–4.2 kg CO_2_eq/kg of live weight at time of slaughter) and lastly, beef cattle (16.27–10.43 kg CO_2_eq/kg of live weight), and sheep (11.42–13.24 kg CO_2_eq/kg of live weight) with a similar level. When comparing the farms with the same species, the farm that does the fattening of calves within the farm itself reveals lower CF levels than the farm selling weaned claves. The same is the case with sheep farms, the farm selling sheep of heavier weights (23 kg) reveals lower CF levels than the farm selling them at 18.5 kg.

If we analyze the group of GHG, the total emissions can be classified in two according to origin: total emissions deriving from the farm and total emissions deriving from the inputs.

In the organic farms under study, the majority of the emissions originate in the farm itself, although they can vary subject to species. In the beef cattle and sheep farms, which are dedicated to meat production, the farm management itself produces over 90% of the emissions. Therefore, the emissions on account of inputs are lower than 10%. However, in the semi-extensive goat and Iberian pig farms, the sourcing of off-farm fodder implies that GHG emissions originating within the farm are 65%, which is a lower value than those of ruminants farms. Whereas, the proportion of the emissions originated in the purchase of inputs, which include mainly the purchase of animal feed, increase.

Within the total farm emissions, the GHG emissions deriving from enteric fermentation in ruminants farms vary between 79.9% and 42.9% of the total emissions, and it is associated to the extensification of these systems and the diet of the animals based on grazing. On the other hand, when we talk about monogastric animals such as pigs, the emissions from CH_4_ enteric fermentation decrease considerably, going down to percentages such as 3.4% and 3.9% for extensive Iberian pig farms.

On the other hand, in pig farms, unlike in ruminants farms, the majority of the emissions derive from manure management, specifically, from manure management direct N_2_O, which yields 36.5% and 43%, respectively.

Soil management and the resulting N_2_O direct and indirect emissions have also been assessed. For the purposes of this analysis, we must take into account that all farms are organic and the production systems are adapted to each species, even when they have common features. The most important feature to take into account is that all the animals spend 90% of their time grazing and therefore they deposit their dung directly on the ground. The results, i.e., Table 4, reveal certain differences between species and their management types. The estimation of (total) GHG on the soil is between 4.73 kg CO_2_eq/FU (30.3%) and 0.21 kg CO_2_eq/FU. These GHG emissions deriving from soil management are mostly due to direct N_2_O, as the quantities calculated for indirect emissions were minimal because there is no manure accumulation.

In terms of the inputs brought into the farms, Table 4 includes both the fuel generated and the fuel consumed, electricity, purchase of livestock feed for each species, age, and type of animal. The emissions deriving from these inputs create major differences between species, in the same way they did for CH_4_ emissions deriving from enteric fermentation. The beef cattle and sheep farms included in this paper reveal values between 2.61 and 9.4 for GHG percentages attributed to off-farm emissions. On the other hand, these acquire importance in the pig and semi-extensive goat farms, especially in terms of the purchase of animal feed (21%–37.7%), thus indicating that farm self-sufficiency based on grazing or self-production of feed is essential and the purchase of feed should be limited. Figure 1 shows the distribution of the carbon footprint components (emissions indicated in kg of CO_2_eq/FU) for each type of farm under study and for all the farms.

### 3.3. Carbon Sequestration

Table 5 shows the results of carbon sequestration relating to the farms acting as GHG sinks.

Carbon sequestration in farm soils has the potential to compensate the emissions deriving from the production systems based on grazing [51]. Therefore, the extensive farms or farm businesses under analysis in this paper are situated on lands with the capacity of fixing GHG emissions in the form of vegetable waste and organic nitrogen. Additionally, the biomass waste remaining in the soil and shock-absorbing the CO_2_ emissions also contribute to restore the soil and to the production of pasturelands. This capacity to shock-absorb emissions is also due to the N to C transformation process occurring when animals deposit their dung while they are grazing and when manure is added. Additionally, authors such as Conant et al., Soussana et al., Byrne et al., and Jaksic et al. [52,53,54,55] in their papers already suggested that, apart from becoming important carbon sinks, soils with permanent pasturelands can also have a major role in C sequestration, particularly when improved grazing strategies are adopted. Veysset et al. [56] stated that should carbon sequestration be taken into account, the compensation percentages would become 40%–70% of the total GHG emissions from the grazing-based systems. Soussana et al. [57] conclude that it is likely for pasturelands in Europe to act as large sinks for the atmospheric CO_2_, which would reduce the CF of milk. However, the paper written by Beauchemin et al. [58] concludes that there is still great uncertainty as to the available estimations, and further research is required before the quantification of the amount attributed to 1 kg of milk can be attained.

Table 5 shows the results of carbon sequestration in the seven farms under study. Such results are expressed as the equivalent total CO_2_ fixed by hectare and by functional unit (FU = kg of meat or kg of milk) and result from the addition of the soil C-sequestration value (pastures and crops) and the N deposited by animals (manure and pasture). They include the total kg of fixed CO_2_ in the pastures and the crops and the total kg of fixed CO_2_ derived from the N deposited through manure and while grazing. The CO_2_ equivalent deriving from the pastures and the crops is obtained from the calculation of the kg of dry matter contained in the farm in question. The estimation of the dry matter of pastures for each farm was obtained according to location and the distribution of the farms. Values between 1000 and 1400 kg on average per hectare were calculated for the various farms [59,60]. N is calculated through the dung depositing of animals, allocating this value between the value that is fixed through the spreading of manure and the accumulated value while grazing. This calculation is performed for each age group in the farm, with the deposited N being different according to age and type of animal.

The carbon input to the soil is from above and below ground grazing land and crop residues (assuming a C content of 45% of dry matter). Table 5 shows all major C inputs each year: C inputs from crop residues and manure. The amount of manure and N excreta per animal per year is based on national data [22,23]. The C:N ratio of the submerged manure was 13.4. However, the current methodology does not allow further adjustments to be made to the soil management as there are no data in the literature on which it can rely.

The final result in terms of sink storage reveals that an amount of between 419.7 and 576.4 kg of CO_2_eq/ha is stored, which goes to prove the importance of extensive farming, where pastures and animals (their dung) play a key role in the agricultural systems. For example, Figure 2 shows the sequestration % in pasture–crop and by way of excrements in manure–soil and according to species.

Lastly, Figure 3 shows the compensated CF by FU. The positive values represent farm emissions in kg of CO_2_eq/FU, whereas the negative values represent the carbon sequestration in these systems, also in kg of CO_2_eq/FU.

## 4. Discussion

The impact of livestock farm production on the environment is particularly relevant for society. Livestock farming is currently directly associated to climate change, the emission of greenhouse gases, and global warming. However, not all the livestock farming production systems produce and/or compensate to the same extent, as there are extensive livestock farming systems which have a function as carbon sequestration systems and can compensate the amounts of equivalent CO_2_ generated by livestock farms [3] to a great extent.

The scientific literature has seen the number of livestock farming CF studies increase. However, the majority of these papers focus on the study of intensive farms such as meat sheep and beef cattle farms [61]; dairy farms [62,63], and intensive pig farms [64]. Other papers have approached the grazing cattle [65,66,67] and grazing goats [68,69,70] production systems. However, very few papers deal with organic livestock farming [9,71,72] that also includes different species and management systems.

The results and conclusions from these papers are hard to compare due to the various production contexts and the methods used, as well as their definition of a functional unit [10,25]. Additionally, there is a limited number of papers on organic farming.

In this study, it has been identified that, organic systems in extensive conditions, the result of CF per unit of product is lower than in other conventional systems. The beef cattle species for the production of organic meat reveals two results: 16.27 kg CO_2_eq for farms selling weaned animals weighing 220 kg of live weight and 10.23 kg CO_2_eq for farms selling fattening animals weighing 450–550 kg of live weight. As can be seen, in farms where animals are fattened, the final CF decreases, although the cycle might be longer than that of the farms selling animals at weaning age. Other papers such as that of Cardoso et al. [65] show results that coincide with the results obtained in this paper, registering lower CF values per kg of live calf sold with the more intensive farming systems. In the same way, the debate is open by other papers indicating that there is a direct association between intensification and lower CF per product unit in farms [9,39]. In this sense, it would become necessary to standardize the functional unit, the system limits, and the allocation method, as well as incorporating carbon sequestration to these studies.

With regards to organic sheep farms producing meat, the emissions derived from enteric fermentation account for 75% to 97.4% of the total emissions. This result seems reasonable, as thanks to feed management in these farms, which is based on extensification and self-sufficiency, and given that the sheep are not finished here, off-farm GHG emissions contribute to the CF value in a lesser proportion. The results found in this piece of research are lower than those drawn from others in similar conditions with non-organic farms such as those of Ripoll-Bosch et al. [73] which vary between 19.5 and 25.9 kg CO_2_eq/kg of live weight sheep in the north of Spain. Other pieces of research such as that of Dougherty et al. [14] show results more in consonance with the results of our research, concluding with a CF of 10.9 to 17.9 kg of CO_2_eq/kg of sheep in the market (sold). In the same way, the comparison of the various pieces of research on the results of farm CF is very sensitive and depends on the method of analysis being used and the way results are presented, either by weight, financial value, or by area, such as Wiedemann et al. [74] states in his paper on the sheep production in the United Kingdom, Australia, and New Zealand.

Semi-extensive organic dairy goat farms register lower values in kg of CO_2_eq/liter of FPCM than the values reported in the literature. Gutiérrez-Peña et al. [10] registered a total amount of emissions of 3.17 ± 0.41, 2.22 ± 0.13, 2.29 ± 0.17 kg CO_2_eq kg^−1^ FPCM1 for the tree types of farms under analysis or 1.88 ± 0.24, 1.31 ± 0.08, 1.36 ± 0.10 kg CO_2_eq kg^−1^ FPCM_2,_ which were more in line with the results found in this paper. These figures do not take into account the total kg CO_2_eq/FPCM that the sequestrated carbon values in our study does. Other papers such as that of Patra [75] allocate 2.54 kg CO_2_eq kg^−1^ to the CF of farms in India. Robertson et al. [69] reveal the average CF they found was 0.90 kg of CO_2_eq/kg of FPCM (without carbon sequestration) lower than that found through our research, but 8.78 t of CO_2_eq/ha, which is substantially above 0.518 t of CO_2_eq/ha calculated for our organic farm, although they state that the CF of the farms under study decreases as the farms become less intensive, with no CF data being provided for dairy goats in organic farms.

It is hard to compare the results obtained for the Iberian montanera pigs, whose final feed is based on pastures and acorn (ripe fruit of the *Quercus* spp.). This feature that is so inherent to the dehesa is the one differentiating these systems from those in the research available on pig’s CF, which are intensive systems, such as that of Arrieta and González [64] who found a CF value of 5.14 and 5.17 ton CO_2_eq/ton LW in paddocks and of 6.06 and 6.04 ton CO_2_eq/ton LW in confinement. Other papers such as that of Bava et al. [76] in Italy, found that for traditional ham-producing intensive pig farms, the CF calculated for six farms yielded an average of 4.25 ± 1.03 kg CO_2_eq/kg of live weight. The results reported in this paper take into account the GHG emissions attributed to soy and its transport. The protein that soy adds to the pig’s diet is essential, but the production of soy is limited in Europe, hence requiring importation from third countries, mainly America and China. In the paper written by Wiedemann et al. [77,78] in Australia, the average calculated was 2.1 to 4.5 kg CO_2_eq/kg of LW.

In organic livestock farming, according to the standards (EU) Regulation 834/2007; (EU) Regulation 2018/848), “the livestock shall have permanent access to open air areas, preferably pasture, whenever the weather conditions and the state of the ground allow”, with the maximum number of animals per ha being limited (2 LU/ha). Nevertheless, even in compliance with this maximum limit, the degree of extensification of organic farms varies to a great extent subject to the production systems and farm dedication. Not only can this variability be seen at the European level due to its large heterogeneity, but also at a smaller scale (regional or local) such as is the case of dehesas. Organic farms in dehesas are highly extensificated with livestock stocking rates significantly below the limits established by the standard (between 0.18 and 0.6 LU/ha). The maintenance of these stoking rates is considered as a sustainability factor [79], given that adequate livestock stocking rates contribute to the ecologic stability of the system, as they prevent shrub invasion (as is the case with under grazing; [80] and the degradation and erosion of the land (as is the case with overgrazing) [81].

But the maintenance of these stocking rates also allows for adequate carbon sequestration by the soil, and its quantification is particularly relevant within the current context of fight against climate change. In the farms under study, in the case of ruminants, the emissions are compensated in 35% to 89%, and they are even compensated in 100% of the GHG emissions in the case of dairy goat farms. In the case of the Iberian pigs, the carbon sequestration exceeds the emissions both in farms dealing with the full cycle and fattening montanera farms. These results differ from other papers such as that of Alemu et al. [66] which included soil C-sequestration but only saw a reduction of the greenhouse gases emission balance in the farm by 12%–25%, with stocking rates ranging 1.2 to 2.5 cow-calf pair/ha.

Maintenance of livestock at stocking rates that are adapted to the productive capacity of the pastures on which they live, reducing the entry of off-farm feed and with capacity to sequestrate carbon, makes organic farms in dehesas a model to follow from the environmental viewpoint, differentiating it from models that pose a threat to the environment. This is the reason why institutions, especially in Europe, must be prepared to discern between the systems that need to be protected and promoted from those that do not have a positive impact.

Currently, the key elements of the post-2020 Common Agricultural Policy (CAP) reform are under debate. The environmental and climate-related aspects are at the center of the debate, as became clear at the Agriculture and Fisheries Council of 15 July 2019 (CAP Progress report 2019), where the delegates highlighted the importance of allowing the member states to have the needs of the locals into account when it comes to applying environmental and climate-related requirements. The debate is focusing on the redefinition of the role of the farmers in climate action and, in particular, in the capture of the soil’s carbon for the purposes of improving its structure and quality, which helps agriculture adapt to climate change.

In the currently effective CAP (EU Regulation 1307/2013, EU Delegated Regulation 639/2014, EU Implementing Regulation 641/2014), there is no standard to regulate or propose specific requirements in relation to soil’s carbon content. However, in its Greening section, some requirements are indirectly proposed for the protection of soil’s carbon, such as the regulations relating to the proportion of the permanent pastures compared to the total declared farming area. We must take into account that the soil’s carbon sequestration is a complex issue and that it is necessary to improve the methods of measurement of carbon, increase research, and put it into practice, relying on innovation that allow for the quantification of the extent to which the CAP contributes to increase those amounts of carbon. The post-2020 CAP reform is an opportunity for the member estates to support carbon retention in the soil by developing national and regional supporting measures that can actually contribute to the fight against climate change.

It is clear that in the present context of CAP debate, the discussions are being focused on the environmental pillar of sustainability, but obviously, the final approach proposed for the other two pillars (economic and social) will also be crucial. It has to be considered that, from an economic point of view, the subsidies (first and second pillar of the CAP) received by organic ruminant farms in the dehesa area represent about 45% of their total income [49]. It cannot be ignored that the livestock production model of these extensive systems is based on small and medium-sized farms, often family-run, with traditional and low-input management. These farms contribute to the settlement of the population in rural areas by facing depopulation, but their dependence on public economic resources is very high and therefore their sustainability may be compromised depending on the economic funds they finally may receive.

In the past, the different models of public policies derived from the implementation of the CAP have had a significant environmental impact on the dehesas. In the period between 1992 and 2000, the model was oriented towards the compensation of income from commercial production activities, resulting in an increase of stocking rates at farm level, intensifying the systems in order to obtain higher levels of income. The intensification led to environmental problems such as lack of tree regeneration and soil degradation and erosion. During the period between 2000 and 2013, the support system known as “decoupling” was established, which had an unequal effect on livestock farming: while cattle maintained their censuses, in the case of sheep and especially goats, the censuses dropped and many farms abandoned their activity, leading to an invasion of scrub, significant changes in the pasture species and landscape alterations [82].

More recently, the CAP 2014–2020 has focused on promoting the development of territories, the efficient use of resources with a view to a sustainable and diverse agricultural sector, paying even more attention to rural areas [49,83]. This approach has led to the maintenance of livestock censuses in dehesa systems and in particular it can be also said that it has been from that moment on that the development of organic farming in Spain has been most notable, with an increase in the number of farms of more than 50% between 2014 and 2018 [84].

The post-2020 CAP that finally takes effect will undoubtedly affect the long-term sustainability of organic livestock farming in dehesas. It is therefore crucial at this time that, specific measures might be included to guarantee the agro-ecological balance of the system, enhancing and compensating economically its environmental functions in order to increase the low income and margins that these farms obtain whilst promoting practices that maintain their ecological stability.

In view of the above, it seems appropriate to consider that the balance of GHG emissions is a good indicator of the environmental sustainability of livestock farming, although not the only one, since in order to quantify the overall sustainability of the dehesa agroecosystem, there are many other environmental, social, and economic indicators to be considered. In this sense, there is research that globally evaluates the sustainability of extensive and organic livestock farms based on a set of indicators of different nature (environmental, economic, and social) [50,79,85]. In a future climate change scenario, the carbon footprint and carbon sequestration are indicators that should be incorporated into a global framework of sustainability and used in a combined way to measure the vulnerabilities of extensive systems to possible effects such as droughts, temperature increase, forest fires, and other extreme weather events that may affect this highly sensitive agroecosystem.

## 5. Conclusions

This paper analyzes the CF of organic livestock farming in seven farms using a life cycle assessment approach, which allowed for the quantification of the GHG balance in the productive process, differentiating it by origin (enteric fermentation, manure management, soil management, feed inputs, and energy use).

On analyzing the origin of the greenhouse gas emissions, our research reveals that enteric fermentation is the major one in ruminants farms. In the case of pigs, however, emissions deriving from manure management are the highest. On the other hand, feed inputs in organic farms are not so relevant as in conventional farms. Organic systems maximize pasture exploitation which in turn contributes to the lesser consumption of off-farm feeds and at the same time, the grazing technique improves the quality of the pasture by increasing soil’s carbon sequestration.

The high capacity of carbon sequestration of the soil in these farming systems of dehesas derives from the large areas of land, which to a great extent compensates for the livestock emissions. In the case of ruminants farms, the emissions are compensated in 35% to 89%, and even in 100% in the case of dairy goats; in the case of Iberian pigs, carbon sequestration levels exceed the emissions. Given these results, particularly highlighting the extensive livestock management system of these ecosystems, we can conclude that the model used by organic livestock farming in the dehesas is a feasible strategy for reducing GHGs from livestock farming.

## Figures and Tables

**Figure 1 animals-10-00162-f001:**
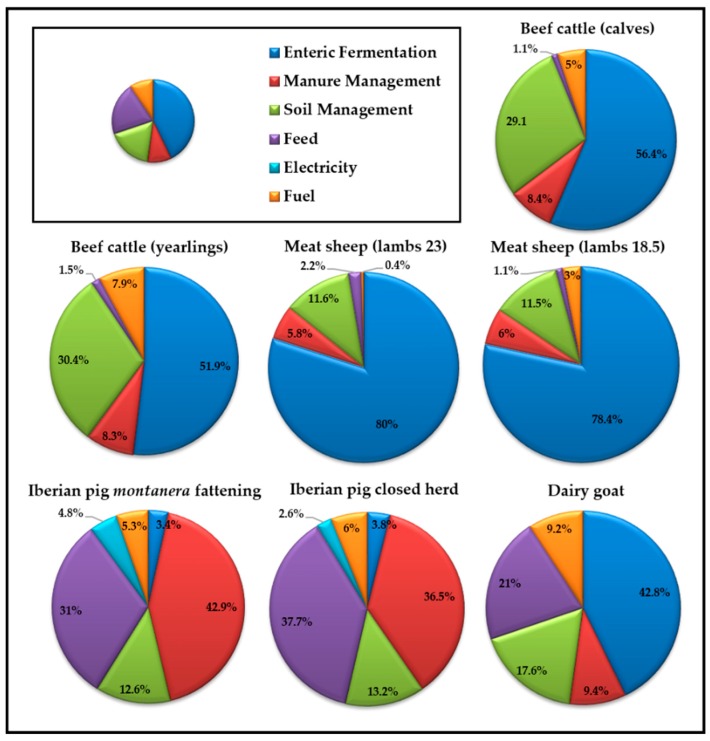
Importance of GHG emission levels by type of farm.

**Figure 2 animals-10-00162-f002:**
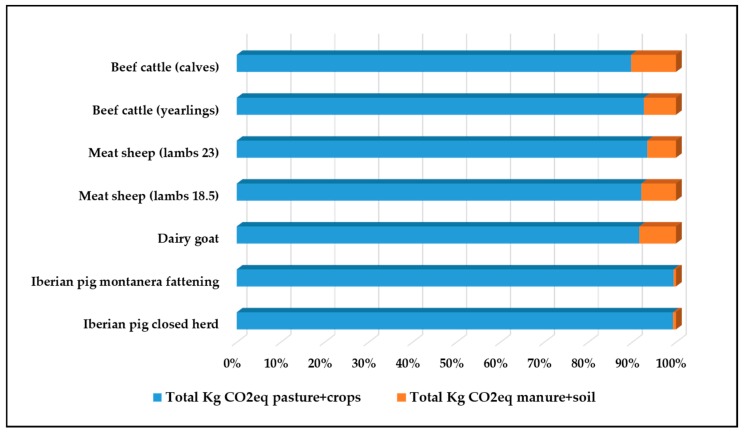
Percentage of carbon in the soil deriving from the vegetable waste and from manure and dung depositing.

**Figure 3 animals-10-00162-f003:**
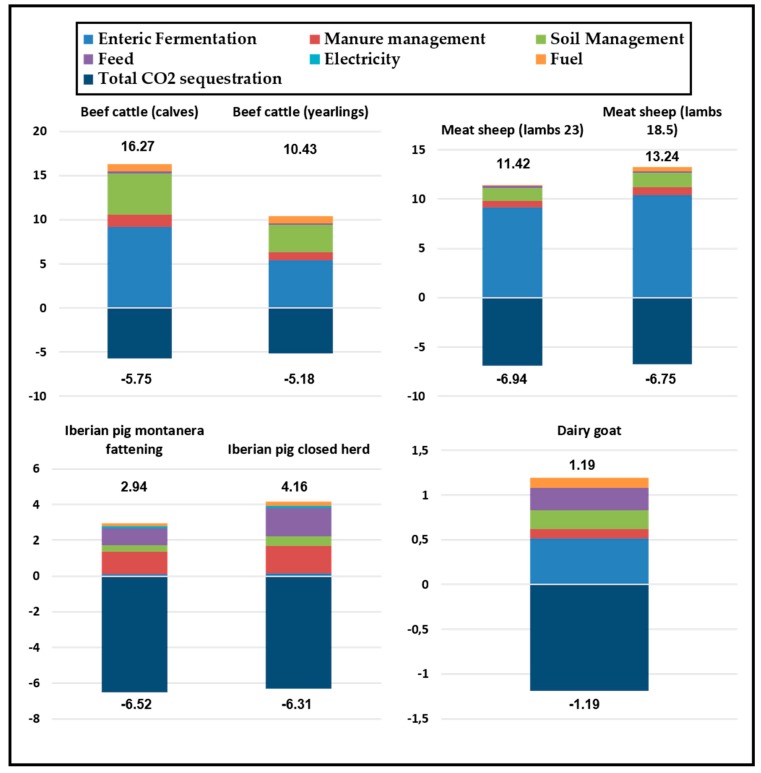
Compensated Carbon Footprint (CF)/functional unit (kg of CO_2_eq/FU).

**Table 1 animals-10-00162-t001:** Main features of the production systems included in the case study.

System Types Description	Photograph
**Beef cattle farm (calves):** Average-size extensive beef cattle farm of 140 ha, with 7.1% of the area dedicated to crops. The expense in feed is approximately 266.7 kg of fodder */reproductive animal and 357.3 kg of concentrates/reproductive animal. The end product of this farm is the sale of weaned claves of approximately 200–250 kg of live weight.	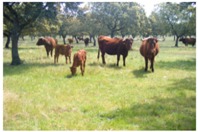
**Beef cattle farm (yearlings):** Small-size extensive beef cattle farm (105 ha) where 2.9% of the area is dedicated to crops. The expense in feed is approximately 136 kg of fodder */reproductive animal and 325.6 kg of concentrates/reproductive animal. The end product of this farm is the sale of finished yearlings with an approximate weight of 500 kg of live weight for males and 400 kg of live weight for females.	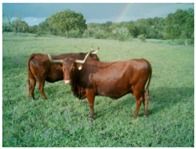
**Meat sheep farm (lambs 23 kg live weight):** Extensive sheep farm of 370 hectares of land and 13.5% of the area dedicated to crops. The expense in feed per sheep is 44.4 kg of fodder */reproductive animal and 103.7 kg of concentrates/reproductive animal and the end product is the sale of sheep of 23 kg of live weight 3 months old.	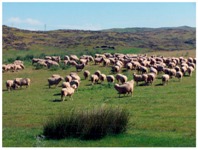
**Meat sheep farm (lambs 18.5 kg live weight):** Extensive sheep farm with a total area of 500 hectares. The area dedicated to crops is 18% a year. The expense in feed is 58.8 kg of fodder */reproductive animal and 85.9 kg of concentrates/reproductive animal. The end product is the sale of sheep of 18.5 kg of live weight approximately from 2 to 2.5 months old.	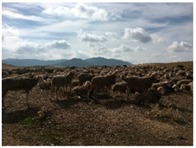
**Dairy goat semi-extensive farm:** Small-size farm (80 ha), with an area of 10% dedicated to crops. The expense in feed per reproductive animal is 72.7 kg of fodder */reproductive animal and 353.8 kg of concentrates/reproductive animal. The end product is the sale of organic milk.	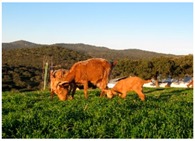
**Iberian pig montanera ^1^ fattening farm:** Iberian pig farm with 50% pure breed pigs, of an area of 300 hectares and 13.3% of the area dedicated to crops. The farm buys its piglets. The end product is the sale of pigs of approximately 160 kg of live weight (age from 14–16 months) which have been fattened on the montanera system.	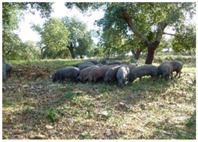
**Iberian pig closed herd farm:** 100% pure Iberian pig farm, with a total area of 230 ha, and 2.2% of the area dedicated to crops. The expense in feed per animal in this farm is 484.4 kg. The end product is the sale of fattening pigs of 40 kg (age from 3–4 months) and montanera pigs of 170 kg in live weight (age from 16–18 months)	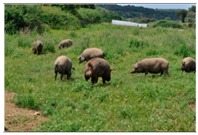

* Fodder refers to straw and hay. ^1^ Montanera is the local name for the free-range fattening of Iberian pigs whereby animals are free to roam in the dehesa and mainly eat acorns (aprox. 10 kg/day) and pasture (aprox. 3–4 kg/day). This covers period from November to February [30].

**Table 2 animals-10-00162-t002:** Emission factors used to quantify greenhouse gas emissions (GHG).

Emission and Source	Type of GHG	Emission Factors	Unit
**On-farm**			
Enteric fermentation	CH_4_	51.06 kg CH_4_/cow a year ^a^	kg CH_4_/year
	CH_4_	7.64 kg CH_4_/sheep a year ^a^	kg CH_4_/year
	CH_4_	5 kg CH_4_/goat a year ^a^	kg CH_4_/year
	CH_4_	2.75 kg CH_4_/breeding pig a year ^a^	kg CH_4_/year
	CH_4_	0.62 kg CH_4_/growing-finishing pig a year ^a^	kg CH_4_/year
**Manure management**			
Manure management CH_4_	CH_4_	6.91 kg CH_4_/cow a year ^b^	kg CH_4_/year
	CH_4_	0.28 kg CH_4_/sheep a year ^b^	kg CH_4_/year
	CH_4_	0.21 kg CH_4_/goat a year ^b^	kg CH_4_/year
	CH_4_	18.76 kg CH_4_/breeding pig a year ^b^	kg CH_4_/year
	CH_4_	7.59 kg CH_4_/growing-finishing pig a year ^b^	kg CH_4_/year
Manure management direct N_2_O	N_2_O	0.005 kg N_2_O eN/kg N solid storage system ^c^	kg N_2_O/year ^d^
Manure management indirect N_2_O	N_2_O	0.01 kg N_2_O eN/volatilized ^c^	kg N_2_O/year ^d^
**Soil management**			
N from urine and dung inputs to grazed soils in Cow (Iberian swine)	N_2_O	0.02 kg N_2_O eN (kg N input)^−1 c^	kg N_2_O/year ^d^
N from urine and dung inputs to grazed soils in Sheep	N_2_O	0.01 kg N_2_O eN (kg N input)^−1 c^	kg N_2_O/year ^d^
N from urine and dung inputs to grazed soils in Goat	N_2_O	0.01 kg N_2_O eN (kg N input)^−1 c^	kg N_2_O/year ^d^
Indirect emissions soil management	N_2_O	0.01 kg N_2_O eN (kg % N volatilized/leaching)^−1 c^	kg N_2_O/year ^d^
**Off-farm**			
Concentrates Meat Cow	CO_2_	0.410 kg CO_2_eq/kg ^e^	kg CO_2_eq/year
Concentrates Meat Calf	CO_2_	0.445 kg CO_2_eq/kg ^e^	kg CO_2_eq/year
Concentrates Meat sheep	CO_2_	0.410 kg CO_2_eq/kg ^e^	kg CO_2_eq/year
Concentrates Meat Lamb	CO_2_	0.410 kg CO_2_eq/kg ^e^	kg CO_2_eq/year
Concentrates Dairy Goat	CO_2_	0.410 kg CO_2_eq/kg ^e^	kg CO_2_eq/year
Concentrates Piglet, 2nd stage feed	CO_2_	0.227 kg CO_2_eq/kg ^e^	kg CO_2_eq/year
Straw	CO_2_	0.100 kg CO_2_eq/kg ^e^	kg CO_2_eq/year
Hay	CO_2_	0.170 kg CO_2_eq/kg ^e^	kg CO_2_eq/year
Wheat	CO_2_	0.335 kg CO_2_eq/kg ^e^	kg CO_2_eq/year
Barley	CO_2_	0.305 kg CO_2_eq/kg ^e^	kg CO_2_eq/year
Pea	CO_2_	0.116 kg CO_2_eq/kg ^e^	kg CO_2_eq/year
Electricity	CO_2_	0.410 kg CO_2_eq/kWh ^f^	kg CO_2_eq/year
Fuel	CO_2_	2.664 kg CO_2_eq/L-Combustion ^e^	kg CO_2_eq/year
	CO_2_	0.320 kg CO_2_eq/L-upstream ^e^	kg CO_2_eq/year

^a^ [23]; ^b^ [22]; ^c^ [37]; ^d^ N_2_OeN*44/28 ¼ N_2_O; and from: ^e^ [32]; ^f^ [33].

**Table 3 animals-10-00162-t003:** Technical indicators of the farms included in this research.

System Types Indicators	Beef Cattle (Calves)	Beef Cattle (Yearlings)	Meat Sheep (Lambs 23)	Meat Sheep (Lambs 18.5)	Dairy Goat	Iberian Pig Montanera Fattening	Iberian Pig Closed Herd
Total Area (ha)	140	105	370	500	80	300	230
Average annual temperature (°C)	16.1	15.7	15.6	15.6	15	16	15.5
Pasture area (%) ^c^	92.9	97.1	86.5	82	81.2	86.7	97.8
Wooded area (%)	100	97.1	-	-	31.2	100	100
Cultivated area (%)	7.1	2.9	13.5	18	18.8	13.3	2.2
No. of reproductive females (average population)	75	25	900	1700	110	-	22
Total stocking rate (LU ^b^/ha)	0.59	0.3	0.44	0.60	0.24	0.18	0.19
No. of weaned animals/reproductive females	0.73	0.68	1.1	1.15	1.7	-	9.2
**Inputs purchased by the farm**							
Total kg of fodder/reproductive animal	266.7	136	44.4	58.8	72.7	-	-
Total kg of concentrates/reproductive animal	357.3	325.6	103.7	85.9	353.8	-	484.4
**Outputs produced by the farm**							
No. of animals sold/reproductive animals	0.73	0.68	1	1.1	1.44	-	9
Liters of milk sold/year	-	-	-	-	30,000	-	-
Weight (kg) average of animals sold	220	* 400/500	23	18.5	9	160	** 40/170
kg of weaned animals	12,100	-	22,770	36,075	2061	-	4000
kg of fattening animals	-	8500	-	-	-	22,400	17,000
Total live weight (kg) produced (FU ^c^)	12,100	8500	22,770	36,075	2061	22,400	21,000

^a^ Pasture area (%): includes with and without trees; ^b^ LU: Livestock Unit; ^c^ FU: Functional Unit (kg of live weight); * 400 kg female and 500 kg male. ** 40 kg for weaner piglets y 150 kg for montanera pigs.

**Table 4 animals-10-00162-t004:** Carbon footprint per functional unit.

	Beef Cattle (Calves)		Beef Cattle (Yearlings)		Meat Sheep (Lambs 23)		Meat Sheep (Lambs 18.5)		Dairy Goat		Iberian Pig Montanera Fattening		Iberian Pig Closed Herd	
GHG Emissions	kg CO_2_eq/kg Product	%	kg CO_2_eq/kg Product	%	kg CO_2_eq/kg Product	%	kg CO_2_eq/kg Product	%	kg CO_2_eq/kg FPMC	%	kg CO_2_eq/kg Product	%	kg CO_2_eq/kg Product	%
**Enteric fermentation CH_4_**	**9.18**	**56.42**	**5.41**	**51.87**	**9.13**	**79.95**	**10.38**	**78.40**	**0.51**	**42.86**	**0.1**	**3.41**	**0.16**	**3.85**
**Manure management**														
CH_4_	1.24	7.62	0.73	7	0.33	2.89	0.38	2.87	0.02	1.68	1.19	40.61	1.46	35.10
Direct N_2_O	0.12	0.74	0.14	1.34	0.33	2.89	0.41	3.10	0.0919	7.72	0.07	3.39	0.06	1.44
Indirect N_2_O	0.00046	0.00	0.00056	0.01	0.0013	0.01	0.0016	0.01	0.0004	0.03	0.0003	0.01	0.0002	0.00
**Total manure management**	**1.36**	**8.36**	**0.87**	**8.35**	**0.66**	**5.79**	**0.79**	**5.98**	**0.112**	**9.44**	**1.26**	**43.01**	**1.52**	**36.54**
**Soil management**														
Direct N_2_O soil	4.15	25.51	2.41	23.11	1.11	9.72	1.27	9.59	0.21	17.65	0.34	11.60	0.50	12.02
Indirect N_2_O soil	0.58	3.56	0.76	7.29	0.22	1.93	0.25	1.89	0	0.00	0.03	1.02	0.05	1.20
**Total soil management**	**4.73**	**29.07**	**3.17**	**30.39**	**1.33**	**11.65**	**1.52**	**11.48**	**0.21**	**17.65**	**0.37**	**12.63**	**0.55**	**13.22**
**Total On-farm Emissions**	**15.27**	**93.86**	**9.45**	**90.61**	**11.12**	**97.38**	**12.69**	**95.86**	**0.83**	**69.94**	**1.73**	**59.05**	**2.07**	**53.61**
**Feeding**														
Concentrate feed cows	0.1	0.61	-		-		-		-		-		-	
Concentrate fattening calves	-		0.16	1.53	-		-		-		-		-	
Concentrate sheep	-		-		0.20	1.75	0.06	0.45	-		-		-	
Concentrate lambs	-		-		0.05	0.44	0.09	0.68	-		-		-	
Concentrate goats	-		-		-		-		0.25	21.01	-		-	
Concentrate growth pigs	-		-		-		-		-		0.91	31.06	-	
Seeds (wheat, barley, vetch)					-		-		-		-		1.57	37.74
Straw	0.08	0.49	-		-		-		-		-		-	
Hay	-		-		-		-		-		-		-	
**Total Feeding**	**0.18**	**1.11**	**0.16**	**1.53**	**0.25**	**2.19**	**0.18**	**1.13**	**0.25**	**21.01**	**0.91**	**31.06**	**1.57**	**37.74**
**Electricity**	**-**		**-**		**-**		**-**		**-**		**0.14**	**4.78**	**0.11**	**2.64**
Fuel														
Production	0.09	0.55	0.09	0.86	0.005	0.04	0.041	0.31	0.012	1.01	0.017	0.58	0.027	0.65
Combustion	0.73	4.49	0.73	7	0.043	0.38	0.36	2.72	0.098	8.24	0.14	4.78	0.23	5.53
**Total fuel**	**0.82**	**5.04**	**0.82**	**7.86**	**0.048**	**0.42**	**0.403**	**3.03**	**0.11**	**9.24**	**0.16**	**5.36**	**0.25**	**6.01**
**Total Off-farm Emissions**	**1**	**6.15**	**0.98**	**9.40**	**0.30**	**2.61**	**0.58**	**4.16**	**0.36**	**30.25**	**1.20**	**41.19**	**1.93**	**46.39**
TOTAL CF kg CO_2_eq/FU	16.27	100	10.43	100	11.42	100	13.24	100	1.19	100	2.94	100	4.16	100

Total kg de CO_2_eq	200,857		90,454		260,314		477,724		40,635		67,267		97,153	
Total kg de CO_2_eq/ha	1434.7		861.5		717.9		974.9		518.3		224.2		422.4	

**Table 5 animals-10-00162-t005:** Carbon sequestration.

Sequestered CO_2_	Beef Cattle (Calves)	Beef Cattle (Yearlings)	Meat Sheep (Lambs 23)	Meat Sheep (Lambs 18.5)	Dairy Goat	Iberian Pig Montanera Fattening	Iberian Pig Closed Herd
C from pasture and crops waste							
**Pasture waste (kg DM) ^a^**	**276,640**	**217,056**	**486,400**	**623,200**	**153,216**	**553,280**	**478,800**
Above ground kg C	32,760	25,704	57,600	73,800	18,144	65,520	56,700
Below ground kg C ^b^	91,728	71,971	161,280	206,640	50,803	183,456	158,760
Crop waste (kg DM)	101,840	30,552	408,880	735,984	71,829	327,104	319,200
Above ground kg C	12,060	3618	48,420	87,156	8506	38,736	37,800
Below ground kg C	33,768	10,130	135,576	244,037	23,817	241,024	235,200
**Total kg CO_2_eq pasture + crops**	**624,492**	**408,553**	**1,477,212**	**2,242,653**	**371,325**	**1,452,634**	**1,316,700**
C from organic N (manure and grazing)							
kg N excreted	5955	2694	8648	16,095	2838	3323	3798
kg C from applied manure	3638	2177	10,571	20,517	5801	216	185
kg C during grazing	15,715	6578	17,534	31,791	4879	1944	2284
**Total kg CO_2_eq manure + soil ^c^**	**70,962**	**32,102**	**103,050**	**191,796**	**33,820**	**7919**	**9052**
Total kg CO_2_eq/farm	695,454	440,655	1,580,262	2,434,450	405,144	1,460,553	1,325,753
Total kg CO_2_eq manure-soil/ha	507	306	279	384	517	135	201
Total kg CO_2_eq farm/ha	4968	4197	4271	4869	5064	3646	3712
**kg CO_2_ pasture + crops sequestration**	**62,449**	**40,855**	**147,721**	**224,265**	**37,132**	**145,263**	**131,670**
**kg CO_2_ manure + soil sequestration**	**7096**	**3210**	**10,305**	**19,180**	**3382**	**792**	**905**
**Total kg CO_2_ sequestration**	**69,545**	**44,066**	**158,026**	**243,445**	**40,514**	**146,055**	**132,575**
**Total CO_2_ sequestration (kg CO_2_eq/FU/year) ^d^**	**5.75**	**5.18**	**6.94**	**6.75**	**1.19**	**6.52**	**6.31**
**Total CO_2_ sequestration (kg CO_2_eq/ha/year^-^)**	**497**	**420**	**427**	**487**	**506**	**487**	**576**
**Compensated CF**							
**Compensated CF/functional unit (kg of CO_2_eq/FU)**	**10.52**	**5.25**	**4.48**	**6.49**	**0**	**−3.58**	**−2.15**
**Compensated CF/ha (kg of CO_2_eq/ha)**	**938**	**442**	**291**	**488**	**12**	**−263**	**−154**

^a^ Pasture waste has been estimated to account for 40% of the total production of pasture, with a C content of 45%; ^b^ According to [37] the default expansion factor for below-ground biomass in semi-arid pasturelands is 2.8; ^c^ The conversion factor for N to C is 13/4 and 44/12 for C to CO_2_; ^d^ Annual C sequestration of 10% is considered.

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
