# Peer review of "Organic Farming as a Strategy to Reduce Carbon Footprint in Dehesa Agroecosystems: A Case Study Comparing Different Livestock Products"

_animals, 2020, doi:10.3390/ani10010162_

Round 1

Reviewer 1 Report

This is an interesting and timely study, which adds further insight to a topic of major concern - how to increase the sustainability of our food production systems. It should be within the interest of the journal's readership. The agricultural systems studied are of interest.

Broadly speaking, the paper has been written in a sufficiently clear way. However, there are a number of points that need to be clarified prior to this being published. The English also needs correcting at times. See specific points below:

Abstract:

L14: delete "the" before "Life Cycle Assessment"

L17-18: delete sentence "Seven different...in total"

Assuming that space permits this, the Simple Summary needs to include a bit more detail about the methodology employed - what was included/excluded in calculations, etc. 

L25: delete "the" before "Life Cycle Assessment"

L34 and L38 (and indeed throughout the manuscript): there is a need for consistency in how 'per' is expressed - is it "/" or "a superscript of "-1"? You use both here (and in other areas of the paper) 

L35 (and elsewhere): need consistency in the number of decimal points for percentage values 

The Abstract doesn't really clearly say what the balance is (sequestration minus emissions) for the production systems.

Introduction:

L48: change to "... such as the soils, water, and biodiversity, being..."

L49: "...and agriculture can highly contribute to..."

L61-62: re-write this last sentence - it is unclear 

L86: spell out CF in full before providing and using the abbreviation (you do this on L101)

L89: insert comma after "Spain", not semi-colon 

There should be greater discussion of the uncertainties and how the decision on where to draw system boundaries will influence the results. 

Materials and Methods:

L109: this is written as if there is only one universal method of LCA, when of course there are many - all of which will give different results. Re-write, accordingly 

L119-120: delete the first two sentences here - it is repetition. Change next sentence to "Data were collected from each of the seven farms by way of..."

Table 1: unclear what the "fodder" refers to - is is grass only, or does it also include e.g. maize, grass silage, hay, straw, etc. - all of which will have different impacts on C emissions due to the inputs needed, etc. 

Table 1: the Beef cattle farm (yearlings) - are the animals sent off for slaughter at that age (presumably so, as it states that they are 'finished yearlings'). This is young (especially so for the females) 

Table 1: I am not sure of the value of including information on the number of employees, family members, etc. - why is this included? Far better to include detail on the number of head of livestock, % tree cover on farms, etc. - factors that will influence both emissions and sequestration 

L135: "...is based on 1 ha of the total hectares of..."

Section 2.4.1: should state what Tier is used for emission factors (given that different sources are used)

L150: "...brought onto the..."

L158: "...shown in Table..."

Table 2 and elsewhere: always use lower-case k for kg

Section 2.5: sequestration also influenced by plant (biomass) growth - not mentioned here (e.g. are there are trees on these seven farms?) 

L179: not sufficiently explained how croplands can be considered as a form of C sequestration - it can obviously have the opposite effect if it involves tillage of soil  

Also no detail as to the soil type/conditions of the farms. E.g., if they are already high in C, this will reduce their capacity to sequester more. If they are of wet nature, this may increase N2O emissions, etc., etc., - not mentioned here  

Results:

L183: insert comma after "paper"

Table 3: numbers of animals and/or stocking density (LSU/ha) would be useful info here, as mentioned earlier 

L200: insert comma after "farms"

L206: "exist"

L207: change "it" to "they"

L215 and 218: spelling of "stoking" should be "stocking"

Table 3: how is "Pasture waste" calculated? The degree of waste will be influenced by management (stocking density, whether it is rotationally-grazed or set-stocked, etc.) - which will in turn influence C inputs from pasture.  This needs better explaining in the accompanying text 

Was a stocking diary kept on the farms for the study year, with weights on/off, etc. noted? This will influence factors such as enteric methane emissions, etc. 

It is also not clear if/how emissions from manure were calculated, or whether it was considered that all manure was returned to soil. 

L322 (and elsewhere): what is the C-balance on these farms? It is the reason why we need the numbers of animals and LSU /ha - it would be good to show what the variation is per ha, as well as per kg of product 

Diuscussion

L349-350: this needs editing as it reads as a 'fact', when the opposite effect could also be true - it depends on the farm/system studied. Re-write, accordingly 

Conclusion

L457-459: this sentence has also been written too strongly, and needs editing, accordingly to 'tone down' the significance of the findings

All in all, a number of points needs addressing above before this is suitable for publishing; but one would hope that the points raised should be easily rectified by the authors   

Author Response

ANSWER TO THE REFEREES – ANIMALS

Manuscript Ref. animals-674362

Title: Organic Farming as a Strategy to Reduce Carbon Footprint in dehesa Agroecosystems: A Case Study Comparing Different Livestock Products.

6-January-2020

Dear Referee,

First of all, we would like to thank you for your review, which allowed us to detect some mistakes in our manuscript that we think have been corrected in this version. In the following paragraphs we give answer to all your questions, either making reference to the correspondent correction in the text, or giving an explanation when we have considered that changes were no needed.

Reviewer 1:

Comments and Suggestions for Authors

This is an interesting and timely study, which adds further insight to a topic of major concern - how to increase the sustainability of our food production systems. It should be within the interest of the journal's readership. The agricultural systems studied are of interest.

Broadly speaking, the paper has been written in a sufficiently clear way. However, there are a number of points that need to be clarified prior to this being published. The English also needs correcting at times. See specific points below:

Abstract:

L14: delete "the" before "Life Cycle Assessment"

Deleted

L17-18: delete sentence "Seven different...in total"

Deleted

Assuming that space permits this, the Simple Summary needs to include a bit more detail about the methodology employed - what was included/excluded in calculations, etc.

This information has been included in the Simple Summary

L25: delete "the" before "Life Cycle Assessment"

Deleted

L34 and L38 (and indeed throughout the manuscript): there is a need for consistency in how 'per' is expressed - is it "/" or "a superscript of "-1"? You use both here (and in other areas of the paper)

Corrected

L35 (and elsewhere): need consistency in the number of decimal points for percentage values

Corrected

The Abstract doesn't really clearly say what the balance is (sequestration minus emissions) for the production systems.

Corrected and added in the Abstract

Introduction:

L48: change to "... such as the soils, water, and biodiversity, being..."

Corrected

L49: "...and agriculture can highly contribute to..."

Corrected

L61-62: re-write this last sentence - it is unclear reviser

Re-written in the text

L86: spell out CF in full before providing and using the abbreviation (you do this on L101)

Corrected

L89: insert comma after "Spain", not semi-colon

Corrected

There should be greater discussion of the uncertainties and how the decision on where to draw system boundaries will influence the results.

A paragraph has been incorporated in the introduction section where it is shown how the level of uncertainty and the limits of the system established may influence the results.

Materials and Methods:

L109: this is written as if there is only one universal method of LCA, when of course there are many - all of which will give different results. Re-write, accordingly

This sentence has been re-written

L119-120: delete the first two sentences here - it is repetition. Change next sentence to "Data were collected from each of the seven farms by way of..."

Corrected and added in the text

Table 1: unclear what the "fodder" refers to - is is grass only, or does it also include e.g. maize, grass silage, hay, straw, etc. - all of which will have different impacts on C emissions due to the inputs needed, etc.

Fodder refers only to straw and hay. This information has been included in the table.

Table 1: the Beef cattle farm (yearlings) - are the animals sent off for slaughter at that age (presumably so, as it states that they are 'finished yearlings'). This is young (especially so for the females)

We understand that this could be an English translation issue. In Spanish, the appropriate name for the finished bovine animal sold for slaughter (produced in this Beef cattle farm) is “añojos” which are animals ranged from 10-18 months old (males and females). We considered that the best word in English for these animals is “yearlings”.

Table 1: I am not sure of the value of including information on the number of employees, family members, etc. - why is this included? Far better to include detail on the number of head of livestock, % tree cover on farms, etc. - factors that will influence both emissions and sequestration

Information on the number of employees and family members has been removed from table 1 and table 3. In addition, number of head of livestock, % tree cover on farms, and other land use information has been included in table 3 as requested bellow.

L135: "...is based on 1 ha of the total hectares of..."

Corrected

Section 2.4.1: should state what Tier is used for emission factors (given that different sources are used).

A paragraph has been incorporated into the text explaining the Tier level according to the different emission factors used.

L150: "...brought onto the..."

Corrected

L158: "...shown in Table..."

Corrected

Table 2 and elsewhere: always use lower-case k for kg

Corrected

Section 2.5: sequestration also influenced by plant (biomass) growth - not mentioned here (e.g. are there are trees on these seven farms?)

A paragraph has been incorporated into the discussion and new information has been collected in the tables.

L179: not sufficiently explained how croplands can be considered as a form of C sequestration - it can obviously have the opposite effect if it involves tillage of soil 

When we talk about crop lands, we generally mean cultivated meadows or rainfed crops for animal feeding. This is when we consider the residues for the carbon fixation of in the soil. This information has been included in the text.

Also no detail as to the soil type/conditions of the farms. E.g., if they are already high in C, this will reduce their capacity to sequester more. If they are of wet nature, this may increase N2O emissions, etc., etc., - not mentioned here 

An explanatory paragraph has been added.

Results:

L183: insert comma after "paper"

Corrected

Table 3: numbers of animals and/or stocking density (LSU/ha) would be useful info here, as mentioned earlier

This information has been included in the table

L200: insert comma after "farms"

Corrected

L206: "exist"

Corrected

L207: change "it" to "they"

Corrected

L215 and 218: spelling of "stoking" should be "stocking"

Corrected

Table 3: how is "Pasture waste" calculated? The degree of waste will be influenced by management (stocking density, whether it is rotationally-grazed or set-stocked, etc.) - which will in turn influence C inputs from pasture.  This needs better explaining in the accompanying text

The carbon input to the soil is from above and below ground grazing land and crop residues (assuming a C content of 45% of dry matter). Table 5 shows all major C inputs each year: C inputs from crop residues and manure. The amount of manure and N excreta per animal per year is based on national data (MAGRAMA 2012 and MITECO 2018, inventories). The C: N ratio of the submerged manure was 13.4. However, the current methodology does not allow further adjustments to be made to the soil management as there are no data in the literature on which it can rely.

This explanation has been included in the text

Was a stocking diary kept on the farms for the study year, with weights on/off, etc. noted? This will influence factors such as enteric methane emissions, etc.

On the farms, the average inventories for the year have been considered.

It is also not clear if/how emissions from manure were calculated, or whether it was considered that all manure was returned to soil.

Emissions from manure were calculated according to allocation portions between manure management and land management (MAGRAMA, 2012). For carbon sequestration, as shown in table 5, excrements left while grazing and manure applied directly to the soil for fertilizing purposes by the farmer are considered.

(esto no se mete)

L322 (and elsewhere): what is the C-balance on these farms? It is the reason why we need the numbers of animals and LSU /ha - it would be good to show what the variation is per ha, as well as per kg of product

Information related to numbers of animals and LSU /ha has been included in table 3.

Discussion

L349-350: this needs editing as it reads as a 'fact', when the opposite effect could also be true - it depends on the farm/system studied. Re-write, accordingly

This sentence has been re-written

Conclusion

L457-459: this sentence has also been written too strongly, and needs editing, accordingly to 'tone down' the significance of the findings

This sentence has been re-written

All in all, a number of points needs addressing above before this is suitable for publishing; but one would hope that the points raised should be easily rectified by the authors  

Reviewer 2 Report

The paper “Organic farming as a strategy to reduce carbon footprint in dehesa agroecosystems: A case study comparing different livestock products” aims at analyse the impact of organic livestock farming in dehesas through the analysis and review of the carbon footprint of seven extensive organic farming systems in various dehesas in the Southwest of Spain”.

The results are clearly presented and the methodological approach used to calculate the Carbon Footprint with LCA is appropriate.

 However, I evidence two main limits of this research

1) the number of farms is very limited: one or two farms per farming system cannot be fully representative of the livestock systems , and the variability can't be considered

I invite the Authors to demonstrate that the selected farms are representative of the livestock systems, and to discuss eventual limits of the sampling design

2) the Authors analysed only Carbon Footprint (CF), whereas for organic farming there are several indicators of environmental footprint that can be used to  discuss their sustainability

I invite the Authors to avoid to compare the CF of different Functional Units; in my opinion it’s more interesting to add other indicators of environmental footprint (land use, cumulative energy demand, feed/food competition, eutrophication… ) and to try to discuss: i) eventual synergies and trade-offs between carbon footprint and other indicators of environmental footprint(sustainability) in organic farms in dehesa agroecosystem; ii) perspectives for the sustainability of organic farms in dehesa agroecosystem

Author Response

ANSWER TO THE REFEREES – ANIMALS

Manuscript Ref. animals-674362

Title: Organic Farming as a Strategy to Reduce Carbon Footprint in dehesa Agroecosystems: A Case Study Comparing Different Livestock Products.

6-January-2020

Dear Referee,

First of all, we would like to thank you for your review, which allowed us to detect some mistakes in our manuscript that we think have been corrected in this version. In the following paragraphs we give answer to all your questions, either making reference to the correspondent correction in the text, or giving an explanation when we have considered that changes were no needed.

Reviewer 2:

Comments and Suggestions for Authors

The paper “Organic farming as a strategy to reduce carbon footprint in dehesa agroecosystems: A case study comparing different livestock products” aims at analyse the impact of organic livestock farming in dehesas through the analysis and review of the carbon footprint of seven extensive organic farming systems in various dehesas in the Southwest of Spain”.

The results are clearly presented and the methodological approach used to calculate the Carbon Footprint with LCA is appropriate.

 However, I evidence two main limits of this research

1) the number of farms is very limited: one or two farms per farming system cannot be fully representative of the livestock systems, and the variability can't be considered

I invite the Authors to demonstrate that the selected farms are representative of the livestock systems, and to discuss eventual limits of the sampling design

This research is bases on a case studies, methodology developed by Yin (1984) [1] on his work titled "Case Study Research: Design and Methods" and it is mainly characterized by an intensive approach to an object of study or unit. It is used for the description of real situations and is applicable, for example, to problems related to the management of enterprises, being in the case of this research the livestock enterprise as the unit of study.

The farming system under study in this paper can be considered unique: an agro-ecosystem grazed by different livestock species under extensive conditions and giving rise to different products depending on the management that the owners of the farms decide to adopt. All these farms are management units subject to the same soil, climate and socio-political conditions located in the Spanish region of Extremadura, an administrative unit of governance.

The selection of seven organic farms for in-depth study has been considered as the appropriate method for achieving the objectives of the study, since each of them is characteristic of a representative management system in the region of Extremadura (the regional area on which the study is focused. It should be mentioned that the number of organic farms in the region is very low, in fact, in the case of organic pig and goat farms, the region only has three farms registered in 2017 and they were all included in the study.

In the recent literature there are numerous studies that use the case study approach for the analysis of livestock farm management from both environmental and technical-economic approaches, for example, Bernués et al. [2] study the environmental impact and ecosystem services of sheep in Spain, Vellenga et al. [3] compare the use of conventional and organic beef cattle water, and Eldesouky et al. [4] analyze the carbon footprint in dehesa farms in Spain. Works with a technical-economic bias are for example those of Neira et al. [5], Asai et al. [6] and Regan et al. [7].

This information has been included in the manuscript in order to adequately justify the selection of the case studies.

2) the Authors analysed only Carbon Footprint (CF), whereas for organic farming there are several indicators of environmental footprint that can be used to discuss their sustainability

I invite the Authors to avoid to compare the CF of different Functional Units; in my opinion it’s more interesting to add other indicators of environmental footprint (land use, cumulative energy demand, feed/food competition, eutrophication… ) and to try to discuss: i) eventual synergies and trade-offs between carbon footprint and other indicators of environmental footprint(sustainability) in organic farms in dehesa agroecosystem; ii) perspectives for the sustainability of organic farms in dehesa agroecosystem

We very much agree with your assessment. We are aware that the sustainability of a system cannot be addressed exclusively by studying a parameter such as the carbon footprint by analysing the values of different functional units. However, that is the purpose of the presented research, in fact, establishing a methodological process according to the characteristics of this agroecosystem is one of the main objectives since, although databases and software are currently available in the market to make this analysis, they do not allow to adjust in a specific way the study parameters at the farm level or allowing the researcher to check all the methodological process. In addition, another of the objectives pursued was to be able to establish the differences in GHG emissions according to the type of product and to identify which components of this GHG (on-far or off-farm) were more relevant and more feasible to reduce. 

The study of sustainability based on a set of indicators has been carried out in previous research [8–10] and for that purpose, economic and social indicators were designed and calculated together with environmental indicators, However, the carbon footprint was not included at that time, since it is a more recent indicator and whose estimation presents an added difficulty given the scarcity of emission factors appropriate to the characteristics of extensive systems. Similarly, carbon sequestration was not included in the above-mentioned studies.

The research presented here for publication in Animals is part of a regional research project. Other tasks that are being developed within the scope of this project are precisely those focused on the economic behaviour of farms and on the combined analysis from an economic-environmental point of view, which could provide more information about the sustainability of the systems according to their productive orientation (cattle-beef-goat-pig).  The results of these other tasks hopefully will be included un another paper, otherwise the length for a single paper would be excessive.

References included in this document:

Yin, R.K. Case study research and applications: Design and methods. Thousand Oaks, CA Sage 1984, 5, 1–53. Bernués, A.; Rodríguez-Ortega, T.; Olaizola, A.M.; Ripoll Bosch, R. Evaluating ecosystem services and disservices of livestock agroecosystems for targeted policy design and management. Grassl. Sci. Eur. 2017, 22, 259–267. Vellenga, L.; Qualitz, G.; Drastig, K. Farm Water Productivity in Conventional and Organic Farming: Case Studies of Cow-Calf Farming Systems in North Germany. Water 2018, 10, 1294. Eldesouky, A.; Mesias, F.J.; Elghannam, A.; Escribano, M. Can extensification compensate livestock greenhouse gas emissions? A study of the carbon footprint in Spanish agroforestry systems. J. Clean. Prod. 2018, 200, 28–38. Neira, D.P.; Montiel, M.S.; Fernández, X.S. Energy indicators for organic livestock production: A case study from Andalusia, Southern Spain. Agroecol. Sustain. Food Syst. 2014, 38, 317–335. Asai, M.; Moraine, M.; Ryschawy, J.; de Wit, J.; Hoshide, A.K.; Martin, G. Critical factors for crop-livestock integration beyond the farm level: A cross-analysis of worldwide case studies. Land use policy 2018, 73, 184–194. Regan, J.T.; Marton, S.; Barrantes, O.; Ruane, E.; Hanegraaf, M.; Berland, J.; Korevaar, H.; Pellerin, S.; Nesme, T. Does the recoupling of dairy and crop production via cooperation between farms generate environmental benefits? A case-study approach in Europe. Eur. J. Agron. 2017, 82, 342–356. Gaspar, P.; Mesías, F.J.; Escribano, M.; Pulido, F. Sustainability in Spanish extensive farms (Dehesas): An economic and management indicator-based evaluation. Rangel. Ecol. Manag. 2009, 62, 153–162. Franco, J.A.; Gaspar, P.; Mesias, F.J. Economic analysis of scenarios for the sustainability of extensive livestock farming in Spain under the CAP. Ecol. Econ. 2012, 74, 120–129. Escribano, A.J.; Gaspar, P.; Mesías, F.J.; Pulido, A.F.; Escribano, M. A sustainability assessment of organic and conventional beef cattle farms in agroforestry systems: the case of the “dehesa” rangelands. ITEA Inf. Tec. Econ. Agrar. 2014, 110, 343–367.

Round 2

Reviewer 2 Report

The Authors has improved the manuscript according to the comments of reviewer.

I think that the discussion can be improved with a section on the global sustainability of organic livestock systems in dehesa.

Author Response

ANSWER TO THE REFEREES – ANIMALS

Manuscript Ref. animals-674362

Title: Organic Farming as a Strategy to Reduce Carbon Footprint in dehesa Agroecosystems: A Case Study Comparing Different Livestock Products.

14-January-2020

Dear Referee,

First of all, thank you very much for this second revision. We appreciate your remarks since they contribute to improve the manuscript in order to be considered for publication in Animals.

Reviewer 2:

Comments and Suggestions for Authors

The Authors has improved the manuscript according to the comments of reviewer.

I think that the discussion can be improved with a section on the global sustainability of organic livestock systems in dehesa.

New information has been included in the discussion section of the manuscript.
